# A Review: The Prospect of Inhaled Insulin Therapy via Vibrating Mesh Technology to Treat Diabetes

**DOI:** 10.3390/ijerph17165795

**Published:** 2020-08-10

**Authors:** Seán M. Cunningham, David A. Tanner

**Affiliations:** School of Engineering, Bernal Institute, University of Limerick, Limerick V94 T9PX, Ireland; Sean.Cunningham@ul.ie

**Keywords:** diabetes, inhaled insulin, vibrating mesh technology

## Abstract

*Background*: Inhaled insulin has proven to be viable and, in some aspects, a more effective alternative to subcutaneous insulin. Past and present insulin inhaler devices have not found clinical or commercial success. Insulin inhalers create a dry powder or soft mist insulin aerosol, which does not provide the required uniform particle size or aerosol volume for deep lung deposition. *Methods*: The primary focus of this review is to investigate the potential treatment of diabetes with a wet insulin aerosol. Vibrating mesh nebulisers allow the passive inhalation of a fine wet mist aerosol for the administration of drugs to the pulmonary system in higher volumes than other small-volume nebulisers. *Results*: At present, there is a significant focus on vibrating mesh nebulisers from the pharmaceutical and biomedical industries for the systemic administration of pharmaceuticals for non-traditional applications such as vaccines or the treatment of diabetes. Systemic drug administration using vibrating mesh nebulisers leads to faster-acting pharmaceuticals with a reduction in drug latency. *Conclusions*: Systemic conditions such as diabetes, require the innovative development of custom vibrating mesh devices to provide the desired flow rates and droplet size for effective inhaled insulin administration.

## 1. Aerosol Therapy

Inhaled aerosol therapy is one of the oldest methods of drug delivery, first documented in approximately 1500 B.C. [1]. An aerosol consists of two phases, a continuous gaseous phase and a discontinuous phase of individual particles that are either solid or liquid [2]. Traditionally inhaled aerosol therapy has been used for topical applications where the aerosol is administered directly to the affected anatomical systems [3]. In the case of topical aerosol therapy, the drug is inhaled and applied directly to the respiratory system for medical conditions such as asthma and tuberculosis.

Only in recent decades has pulmonary drug delivery for systemic applications (i.e., conditions outside the respiratory system) been viable, due to increased efficacy of aerosol therapy [4]. Systemic aerosol therapy relies on the efficiency of deep lung deposition of sufficient drug dosage to the alveolus [5]. Deposition of aerosol into the alveolus in the respiratory airways (i.e., the active region of the lung) increases the bioavailability of the prescribed drug for either topical applications in the lung or systemic applications by entering the circulatory system via the pulmonary system. Effective aerosol delivery to the pulmonary system gives expansive and novel applications to aerosol therapy, such as inhaled insulin to treat diabetes [6].

The pharmaceutical industry is engaged in the prospect of aerosol therapies for systemic applications [4]. The large surface area of the lungs allows for quicker absorption of the drug into the bloodstream when compared to subcutaneous drug delivery [5]. Faster absorption time reduces drug latency, which allows drugs to become fast-acting. Reduced drug latency is crucial for specific applications such as diabetes, achieving better glycaemic control. Aerosol therapy as an alternative drug delivery method is much less invasive compared to the current subcutaneous administration of insulin. Aerosol therapy allows for passive drug delivery, which is also crucial for neonatal intensive care, paediatrics, and intensive care units.

### 1.1. Aerosol Droplet Size

Nasal passages have evolved to protect the lower airways from constant exposure to airborne pathogens and particles, specifically, particles larger than 3–10 µm, which are efficiently filtered out and trapped by the mucus blanket [7]. However, the effective size of the droplets in an aerosol should be 2–5 µm, which results in about 20% efficiency for deep lung deposition [8]. Above 5 µm, the droplets fall out of the flow and diffuse into the upper airways. Below 2 µm, the particles do not experience inertial effects to follow the inhaled airflow, which results in the smaller droplets either evaporating or being exhaled [9]. From literature, it is clear that the fluid properties (surface tension and density) and orifice radius directly affect the droplet size, within an aerosol [10]. The properties of the fluid cannot be altered, as the fluid must be suitable to carry medication for aerosolisation. Therefore, to reduce the droplet diameter and size, the radius of the orifice must be reduced [11].

### 1.2. Aerosol Deposition Mechanisms

For physiological effects to occur, inhaled aerosol droplets must first deposit within the airways [12]. A large number of factors contribute to particle deposition within the airways, such as size, shape and density, airway flow velocity and volume, interpatient physiological variations, and pause time between inspiration and expiration. These factors contribute to total aerosol deposition within the airways due to three main deposition mechanisms: impaction, sedimentation, and diffusion. Impaction of aerosol droplets upon airway surfaces is influenced by aerosol droplet size, density, and velocity. As a result of inertial forces, it is most likely to occur in the upper airways characterised by high aerosol droplet velocities and drastic changes in the airflow direction. Inertial forces are most influential upon aerosol droplets with a diameter greater than 5 µm [9]. Sedimentation of aerosol droplets within the airways is dependent upon droplet mass. Aerosol droplets will be subject to sedimentation if the product of their settling velocity and residence time is greater than the distance required for the contact of the surface airway. Sedimentation typically influences droplets between 0.5 and 2 µm [12]. Sedimentation of larger droplets occurs due to inertial effects at the 90° transition between the mouth and throat [12].

The diffusion of droplets is influenced by the residence time of the droplets within the airways. Residence times within the respiratory tract may be sub-second, and the final dynamic effects influencing aerosol droplet deposition within the airways are a result of local aerodynamics, which are determined by local morphology of the airways and airflow changes throughout the breathing cycle. During exhalation, the nasal valve acts as a brake. This braking allows more time for the gas exchange in the alveoli and the retention of fluid and heat from the warm saturated exhaled air [7]. Droplets less than 0.2 µm are subject to deposition via diffusion based on Brownian motion. A reduction in droplet size and increase in residence time increases the probability of the droplet to deposit through diffusion. Furthermore, breath-holding increases deposition via this mechanism [12].

### 1.3. Anatomy of the Airways

The anatomy of the airways comprises two distinct regions: The conducting airways, also known as the upper airways, beginning at the mouth/nose, and encompasses the trachea, bronchi, bronchioles, and terminal bronchioles. Studies have demonstrated that the trachea presents >90% of airway transport resistance, which limits the potential for deep-lung drug delivery [13]. Within the conducting airways, no gas exchange takes place; its primary purpose is to transport gas to the respiratory airways [14]. A secondary function of the conducting airways is to ensure that the inspired gasses are humidified and heated to provide the alveoli with air identical to the pre-existing environment. The danger associated with cold air inspiration consists in loss of body heat (i.e., a drop in the core temperature) due to heat transfer between the body and respired air, as well as the water loss due to humid-air expiration [12,15]. Accurate humidity of respiratory gasses is crucial to ensure proper functionality of the airways. Improper humidity leads to extensive dehydration and loss of body weight, as functional impairment of the mucociliary escalator occurs rapidly [16]. The mucociliary escalator is a mucus barrier that lines the airway tract and fights infection. The potential damage caused by dry air inspiration may cause the destruction of cilia contributing to the damage of mucous glands and disorganisation of basement membrane [15] and resulting in respiratory issues, leading to chronic cough. Furthermore, over-humidified air poses a danger, i.e., water intoxication, with the final effects being analogous to those listed for dehumidified air in the opposing direction [17]. Improperly functioning conducting airways lead to impaired respiration and bronchoconstriction, which, in turn, leads to inefficient drug delivery to the lung or potentially impairs pulmonary absorption ability [15].

Distal to the terminal bronchioles in the conducting airways lie the respiratory airways, which consist of the respiratory bronchioles, alveolar ducts, and alveolar sacs, and collectively form the lung. The primary function of this region is a gaseous exchange, which may take place throughout the listed bifurcations [12]. The respiratory airways are suited to gaseous exchange due to inherent physical characteristics. The alveolar ducts are typically 1 mm in length formed via connected groups of alveoli, polyhedral chambers with an average diameter of 250 µm characterised by a 0.1–0.4 µm epithelium and 70 nm liquid lining layer [15]. The surface area of the respiratory airways is approximately 102 m^2^, while the conducting airways is a mere 2–3 m^2^. The respiratory airways have much higher contact with inspired gas or aerosol [18]. The thickness of the cell layer, which makes up and lines the respiratory airways, is progressively reduced from 60 µm in the conducting airways to a sub-micron thickness in the alveoli [19]. The fluid layer at the cell surface in the respiratory airways decreases from 8 µm to approximately 70 nm, which directly correlates with a decrease in cell thickness [19].

## 2. Objectives

To review the failures and shortcomings of current inhaled insulin therapy.To review the suitability of vibrating mesh nebulisers to administer inhaled insulin effectively.To define the aerosol flow rate and droplet size to treat type-II diabetes effectively using a vibrating mesh nebulisers.

## 3. Insulin Therapy

Intensive insulin treatment for type-I diabetes involves multiple daily subcutaneous insulin injections (3 to 5 per day) [20,21], with both long-acting basal insulin and short-acting prandial insulin [22,23,24]. Current strategies of subcutaneous insulin administration do not mimic this first-pass effect (a phenomenon where the insulin concentration is significantly reduced due to absorption in the liver before the insulin can reach systemic circulation) of insulin on hepatic glucose control [25,26]. Patients often resist transitioning to subcutaneous insulin administration due to fears [27,28,29] and concerns around the need to accurately correlate carbohydrate intake with insulin administration, as well as competency with a hypodermic needle [30,31]. Due to these concerns, delayed intensification of insulin therapy for people with type-I diabetes occurs, and adherence to injection regimens may be suboptimal [22].

Continuous subcutaneous insulin infusion (CSII) pumps can monitor, administer, and control insulin administration. Studies comparing traditional injected subcutaneous insulin to CSII demonstrated that patients using the implantable peritoneal insulin pump had a reduced average glycated haemoglobin level (HbA1c) and spent more time in the euglycaemic range and less time in the hyperglycaemic range [31,32]. However, cost and concerns over the risk of peritoneal infection and implantation-site complications limit the application of CSII [32]. Insulin pumps are primarily used for type-I diabetes, as the patient is reliant upon continuous insulin administration [32]; continuous insulin infusion is not required by people with type-II diabetes [31]. Thus, there is a desire to develop new forms of delivery devices to administer insulin to treat type-II diabetes [27].

Innovative and novel means of insulin delivery could significantly increase the use of insulin as a method for controlling blood glucose levels. Inhaled insulin administration could substantially improve the quality of life for people with diabetes by overcoming the burdens and perceptions associated with the traditionally administered subcutaneous insulin. Oral administration of polypeptide hormones, such as insulin, results in loss of biopotency owing to a breakdown in the stomach. Several parenteral administration routes of insulin administration other than subcutaneous and inter-venous routes have been studied including transdermal, buccal, nasal, and pulmonary delivery among these. The lung provides an attractive option for insulin therapy, given its accessibility and extensive alveolar capillary network for drug depositions [27]. Pulmonary delivery of insulin, which was first trialled in 1924, has been an area of active research and interest. Soon after the discovery of insulin by Banting and Best in the 1920s [33], studies commenced into the possibility of effectively delivering insulin to the pulmonary system. These initial studies outlined that blood glucose (HbA1c) reduced in response to inhaled insulin [33]. Studies demonstrated that inhaled insulin controlled blood glucose (HbA1c) with comparable results to subcutaneous insulin in 6 children with type-I diabetes [32]. However, the bioavailability of the inhaled insulin was substantially lower than that of the subcutaneous insulin. This reduced bioavailability of the inhaled insulin was due to the low volume of deep lung deposition of the aerosol [34]. Consequently, the recent development of aerosol delivery devices and further research in particle pharmacology made inhaled insulin more viable [22].

### 3.1. Inhaled Insulin: Devices Development

There have been a wide range of mechanically actuated insulin inhalers that have been developed, all using various technologies but no such nebuliser devices specifically designed for an inhaled form of insulin. Devices capable of delivering insulin particulate to the alveolus have been developed and studied in a variety of clinical protocols [32,33,34,35]. The ideal device should not only be capable of consistently delivering insulin to achieve optimal glycaemic control but also should be convenient for patients—both portable and user-friendly [22]. Over the past 30 years, there have been numerous attempts by several companies to develop an inhaled insulin system for domestic patient use. A variety of inhaler devices were developed, which rely on various aerosol mechanics and formulations of insulin that can be inhaled—such as liquid versus lyophilised powder insulin. The performance of the devices differs significantly concerning droplet size, mechanism of insulin release, and regulation of insulin administration. The effectiveness of these devices differs with varying bioavailability of inhaled insulin for each of the devices. Studies have shown that the bioavailability of dry powder insulin is ≈10%, while that of wet aerosol is more significant at ≈46% with much of the insulin being lost within the delivery device or in the oropharynx or upper airways [34].

#### 3.1.1. AERx iDMS

The AERx Insulin Diabetes Management System (iDMS) was developed in collaboration between Aradigm Corporation and Novo Nordisk. This delivery device uses insulin in the form of pre-prepared liquid blisters. The device has an electronic control that guides the user to inhale the insulin in a reproducible fashion. The AERx iDMS was developed with the ability to download the dosing, frequency of use, and inhalation patterns to aid the physician and patient in monitoring treatment goals and adherence [22]. Performance studies with the AERx iDMS were carried out in patients with type-I diabetes mellitus (T1DM), which demonstrated that there was a more rapid rise in serum insulin in the inhaled group versus traditional subcutaneous insulin [35]. However, this study also highlighted that the intrasubject variability concerning the total insulin exposure was 26% for the inhaled group, indicating the consistent inhalation techniques could play a significant role in the treatment of diabetes [22]. However, when the AERx iDMS system was in phase III trials of Food and Drug Association (FDA) approval, Novo Nordisk elected to discontinue further studies with the system.

#### 3.1.2. Exubera

Exubera was developed through a collaboration of Nektar Therapeutics and Pfizer. In 2006, Exubera was approved by the Food and Drug Association (FDA) and the European Medicines Agency (EMA) for the treatment of both type-I diabetes (T1DM) and type-II diabetes (T2DM). The insulin delivered by the Exubera is in a dry powder form. The powder is contained in pre-packaged blister packets containing 1 or 3 mg of regular human insulin. The inhaled dose is delivered through a mechanical inhaler that is breath actuated, which delivers the equivalent of three units or eight units of subcutaneous short-acting insulin. A study was commissioned comparing fast-acting prandial insulin delivered by Exubera versus traditional subcutaneous insulin delivery [36]. This study was carried out on trial subjects comprising non-smoking healthy adult males. The total insulin exposure was similar for both the Exubera-delivered insulin and subcutaneous insulin. However, the time to maximal insulin concentration (CMax) was more rapid for inhaled Exubera insulin versus traditional subcutaneous insulin, with 55 versus 148 min. Exubera was commercially available for a short time between August 2006 to October 2007 before the manufacturer discontinued it as the new technology failed to gain broad acceptance by patients or clinicians, which led to poor sales [37]. The experience with Exubera represented a significant setback for the field, with most pharmaceutical companies re-evaluating and cancelling their inhaled insulin research and development programs.

#### 3.1.3. AIR

The AIR insulin system is dry powder mechanical insulin inhaler, which was developed in collaboration between Eli Lilly and Co. and Alkermes Inc. This inhaler produces a substantially larger particle size (5–30 µm), which is significantly outside the particle size as required for deep lung deposition (2–5 µm) [8]. Nevertheless, the AIR creates a less dense particle compared to other systems, which makes it possible to deposit into the respiratory airways efficiently [22]. This device has been through extensive testing for phase III FDA approval. However, Eli Lilly and partners have decided to discontinue this project and are not pursuing the development of this product any further [22].

#### 3.1.4. Afrezza

In June 2014, the U.S. Food and Drug Administration (FDA) approved a new formulation of the Afrezza inhaled insulin powder to act as prandial insulin requirements for non-smoking adults with diabetes who are free of pulmonary disease. The new drug-device product became available for clinical use in the United States in February 2015. The Afrezza device consists of a dry powder inhaler (DPI) and a new dry powder formulation of recombinant regular human insulin (Technosphere insulin), which is packaged in a pre-filled cartridge and delivered through a handheld pocket-sized device. Technosphere insulin (TI) is a dry powder recombinant human insulin. This system was developed in collaboration between Mannkind Corp., by developing the recombinant human insulin powder, and Pharmaceutical Discovery Corp., who developed the MedTone DPI device. This system was trialled with a novel approach, as a placebo formulation for inhalation was developed. This allowed for the design of double-blind, placebo-controlled studies with people with type-II diabetes (T2DM) [38]. Studies with healthy volunteers compared Technosphere inhaled insulin and traditional subcutaneous insulin. This study demonstrated that the maximal insulin concentration (CMax) was reached in 15 min by the Technosphere inhaled insulin and in 120 min by the traditional subcutaneous insulin; moreover, CMax was 45% greater with inhaled Technosphere insulin compared to the subcutaneous insulin [36]. This study also established that while the total insulin exposure for inhaled insulin was comparable to that for the subcutaneous insulin, the exposure time was shorter with inhaled insulin, suggesting that the risk of delayed hypoglycaemia may be less with the inhaled insulin formulation [38,39]. For insulin to be administered through the pulmonary system, inhalation devices must provide consistently accurate doses [40,41,42,43,44]. After inhalation and upon contact with the lung surface, Technosphere insulin (TI) particles carrying regular insulin molecules dissolve rapidly at physiologic pH, allowing both excipient and insulin to be rapidly absorbed in the pulmonary system [45,46,47]. TI is nearly completely cleared from the lungs when compared with the Exubera formulation, with 0.3% versus 9% remaining in the lung after 12 h [48]. Due to decreased bioavailability (approximately 25% that of subcutaneous insulin), more insulin must be administered through inhalation than by subcutaneous injection to achieve an equivalent therapeutic response [48].

### 3.2. Clinical Trials: Afrezza Inhalable Insulin

In clinical trials, a reduction in glaciated haemoglobin (A1C) with inhaled Technosphere insulin was 7% less than that of subcutaneous insulin, for both type-II and insulin-dependent type-I [49]. Furthermore, the degree of glycaemic control with TI and the proportion of the trial achieving desired goals was much lower compared with historical trials with traditional subcutaneous insulin [50]. The reduced glycaemic efficacy of TI may be due to suboptimal treatment employed in the pre-marketing clinical trials and raises the question of whether therapy with subcutaneous insulin was applied appropriately in the comparator and truly represented “Standard of Therapy”, or whether the comparison was between inhaled insulin and substandard subcutaneous insulin therapy [39]. Insulin titration regimes to reach pre-specified glycaemic goals were either lacking or were not enforced in the trials with TI, which may have resulted in inadequate optimisation of TI [51]. Had the insulin regimens been adequately titrated, there may have been more significant differences favouring subcutaneous insulin. Another explanation for the modest efficacy of TI might be a “ceiling effect”. Which refers to the property of increasing doses of a given medication having progressively smaller incremental metabolic effects. In contrast with subcutaneous insulin, TI exhibits a dose–response relationship where increasing doses do not result in proportional glucose-lowering effect, which suggests a plateau effect for TI [52,53].

#### 3.2.1. Hypoglycaemia

Hypoglycaemia is a common complication of insulin therapy, including inhaled insulin. In patients with type-II diabetes, severe hypoglycaemic events occurred in 5.1% of patients treated with TI, versus 1.7% hypoglycaemic events treated with placebo (Technosphere powder without insulin), and non-severe hypoglycaemia occurred in 67% versus 30%, respectively [54]. In clinical trials with people with type-I and type-II diabetes, severe hypoglycaemic events were reported. Hypoglycaemic events were less frequent with TI versus with subcutaneous insulin (11.7% of patients treated with TI versus 17.8% with subcutaneous insulin) [55,56,57]. The rate of hypoglycaemia in clinical practice is difficult to predict due to the novelty of using a new delivery device and because dose adjustment with TI has yet to be optimised for finer incremental dosing needed to minimise hypoglycaemia especially in insulin-sensitive patients [58].

#### 3.2.2. Weight Gain

In patients with type-I or type-II diabetes, TI was associated with less weight gain compared with subcutaneous insulin and oral pharmacotherapy (1 versus 1.64 kg) [44]. Less weight gain was also reported with Exubera versus subcutaneous in a pooled analysis of six months of data from five phase III Exubera studies. Exubera resulted in more weight gain compared with metformin or sulphonylureas (net difference of 1.85 kg, 95%) [59,60,61].

#### 3.2.3. Persistent Cough

The most common pulmonary symptom, a dry cough, associated with inhaled insulin was reported in as many as 44% of patients treated with TI. In clinical trials, a cough was reported approximately eight times more frequent with TI, when compared with the active comparator group [57]. The dry cough is predominantly mild, occurs within 10 min of inhalation, and is not associated with changes in pulmonary function tests. The cough is noted early in the treatment course and declines in frequency and severity over time but was still reported by 7% of patients long-term (up to four years), follow-up studies outline the presence of this cough [43,44,58]. The cough is likely explained by transient airways irritation or mild bronchospasm due to the delivery method of a dry powder [52]. A cough is the most common adverse event resulting in discontinuation of inhaled TI and the Afrezza Inhaler [44].

#### 3.2.4. Patient Satisfaction of Afrezza Inhalable Insulin

Studies were undertaken where patients received educational information on the availability of inhaled insulin as a treatment option [62]. Findings demonstrated an increase of approximately three-fold of the proportion of patients who would theoretically choose TI overall. However, in three studies with TI, which lasted over 45 weeks in patients with type-I or type-II diabetes, found there was no difference in the quality of life, overall patient satisfaction, or treatment preference between TI and subcutaneous insulin [43,63,64]. In pre-marketing trials, patients treated with TI were more likely to discontinue participation compared with those treated with an active comparator (29.5% for TI versus 15.3% for subcutaneous insulin or oral anti-diabetic drugs) for both type-I and people with type-II diabetes [43,44,52]. In a two-year trial, almost 50% of patients assigned to TI withdrew, compared with 30% receiving usual care. The most common reasons for withdrawal were patient or clinician decision due to inadequate glycaemic efficacy and treatment-emergent, intolerable adverse events (cough) [44].

#### 3.2.5. Insulin Administration and Dosing

The Afrezza is a drug–device combination product that consists of a handheld, pocket-sized, breath-powered inhaler device that accepts single-use cartridge pre-filled with a dry powder formulation of recombinant regular human insulin (Technosphere insulin) that is optimised for inhalation and rapid absorption through the lungs [54]. Inhaled insulin is administrated at the beginning of a meal. In patients with type-I diabetes, inhaled insulin should be taken with subcutaneous long-acting insulin. In patients with type-II diabetes, inhaled insulin can be used as a monotherapy or in combination with oral agents or subcutaneous longer-acting insulin [54]. Technosphere insulin (TI) is available in colour-coded cartridges of 4 (blue), 8 (green), and 12 (yellow) units as a single inhalation per cartridge [53]. For patients who are insulin naïve, the starting dose is four units with each meal.

As with other insulin formulations, the dose is adjusted based on self-monitoring of blood glucose and A1C assessments. Therefore, requiring specific insulin dosage may be difficult. If a patient needs a dose exceeding 12 units, inhalations using multiple cartridges are necessary (e.g., a patient requiring 16 units with dinner will require two different inhalations of the 8-unit cartridge). Afrezza packages are available in multiple configurations that include a combination of the three colour-coded cartridges; a titration package includes ninety 4-unit and ninety 8-unit cartridges. Each package of Afrezza includes two inhalers. The patient should use one inhaler at a time and replace it after 15 days.

Finally, due to dosing inflexibility relative to subcutaneous insulin, it may be challenging to achieve narrow glucose goals with inhaled insulin preparations. Therefore, having patients replace their current, effective subcutaneous therapy with inhaled insulin may not result in the same degree of glycaemic control. In TI trials, perceived lack of efficacy by patients or clinicians was the most common reason for patients’ withdrawal.

## 4. Review of Inhaled Insulin Delivery Devices

In the second half of the 20th century, four distinct inhaled pharmaceutical delivery systems were developed and used clinically: These are nebulisers, pressurised metered dose inhalers (pMDIs), dry powder inhalers (DPIs), and soft mist inhalers (SMIs) [1,6]. The nebuliser has developed from a simple stream device to jet nebuliser adaptive aerosol delivery systems and more recently, vibrating mesh nebuliser (VMN) [4].

Nebulisers are distinct from inhalers, as nebulisers require an aqueous solution or suspension formulation, an aerosol generation device, and an energy source [6]. With inhalers, the drug solution is already present in the device once received. Unlike these inhalers, the nebuliser device is designed and developed independently from the development of the inhaled pharmaceutical formulation [4]. This is because there is a wide range of liquid drug suspensions and solutions that can be aerosolised by a nebuliser, and a specially designed nebuliser for a specific pharmaceutical product is not viable. The liquid drug suspension system in nebulisers allows for ease of filling and flexible dosing of the pharmaceutical [6].

The generated aerosol from a nebuliser is typically inhaled for between 10 and 20 min based on the nebulising regime and dose required. This extended nebulisation time allows for a higher dose than inhalers. As a nebuliser is unassisted, the patient passively inhales the drug, unlike the required patient compliance for pMDIs, DPIs, and SMIs for adequate drug delivery. The advantages of aerosol drug administration ensure that the patient does not need to be conscious to receive measured dosage. The non-invasive administration method ensures the patient is comfortable, which leads to optimal patient compliance.

Along with these advances in nebuliser design, the microchip has allowed for more considerable advances in nebuliser function in the last 15 years. The developments of nebulisers in recent times have taken the form of many different designs that rely on varying strands of technologies. The two main strands of technology used are mechanically and electrically driven nebulisers [3]. Vibrating mesh nebulisers are the most suited nebuliser design to nebulise insulin most effectively and resulting in the best clinical outcome [65]. Nebulisers have historically been used in hospitals or home aerosol therapy. However, next-generation nebuliser technologies aim to expand applications for portable use for systemic drug delivery, reducing inhalation time and improving the delivered dosing. Nebulisers can readily deliver much larger drug doses than other respiratory drug-delivery devices to the large alveolar surfaces of the lungs for drug absorption [65].

### 4.1. Vibrating Mesh Technology

A significant innovation was made in the nebuliser market around 2005 with the creation of the ultrasonic vibrating mesh nebuliser (VMN) [66,67]. Figure 1 illustrates three different vibrating mesh nebulisers that are currently available on the market.

The VMN has become the standard of nebulisation care in many hospitals [69]. Clinical researchers have established its superior performance and the potential for cost savings in comparison to other nebulisation devices. It has been documented that staff satisfaction significantly increased after switching to the Aerogen Aeroneb vibrating mesh nebuliser device [70]. They also highlighted a potential system-wide annual saving of up to $1.74 million across 105 hospitals. The VMN can provide the patient with up to a nine-fold higher drug dose than a standard small volume nebuliser (SVN) (i.e., Jet nebuliser) during mechanical ventilation [71,72]. The VMN technology consists of an aperture plate with thousands of formed or laser-drilled orifices. The vibrating mesh is in direct contact with the reservoir of liquid/medication. A piezoelectric component expands and contracts when an AC voltage is applied, which causes the aperture plate to vibrate. The medication is placed in a reservoir above the aperture plate. During operation, the movement of the aperture plate creates a micro-pumping action that forces the liquid pharmaceutical through the apertures to produce an aerosol. The aerosol particle size and the flow rate are determined by the exit radius of the orifices hole, which can be altered for different clinical applications. VMN devices are capable of nebulising at a rate of 0.4–0.6 mL·min^−1^, and at the end of the delivery, virtually no drug is left in the medication chamber (low residual volume). These VMN devices can nebulise a wide range of medications including suspensions, proteins, and peptides, this is as long as there is no significant increase in the temperature of the solution, and, hence, there is little risk of denaturing the proteins or peptides or reducing the activity of antibiotics during delivery [4].

#### 4.1.1. Published Clinical Trials: VMN for Topical Applications

Physiological lung dose was studied in neonatal animal models, where 99mTc-DTPA was measured after aerosol inhalation through a ventilator circuit [67]. The Aeroneb Pro demonstrated a 25-fold higher deposition of aerosol in the lungs compared to the standard small volume nebulisers (SVNs) [67]. The superior drug deposition available with Aeroneb nebulisers is associated with minimal residual volume remaining in the device reservoir after nebulisation. Standard SVNs on average leave up to half of the drug behind. This can be quite costly when using more expensive drugs [70]. The standard SVN has a residual volume of 1.1 mL after nebulisation of 3 mL of 99 mTc-DTPA. In contrast, the Aeroneb Pro has a residual volume of 0.1 mL [67]. Lung scintigraphy uses a gamma camera to track gaseous radionuclide xenon radioactive tags in the lung during a ventilation scan. The healthy patient inhaled these radioactive tags through a mouthpiece and determined a percentage lung deposition of 18.3% for a VMN and 7.85% for an SVN. VMNs provide superior therapy within the acute care setting during ventilation [70,71]. In addition to VMNs’ optimal performance, substantial cost savings have also been acknowledged by care providers [72]. VMN aerosol therapy is now available across the acute care setting to all respiratory patients, including those who do not require mechanical ventilator assistance [73].

#### 4.1.2. Published Clinical Trials: The Dance 501 VMN Inhaled Insulin

The Dance 501 is an insulin-specific VMN that was developed by Dance Biopharm under a licensed patent from Aerogen and is based on the current commercial Solo II device [74]. The Aerogen Solo II device was life tested to operate continuously for seven days within an acute care setting. A breath-actuated inhaler could potentially last up to 6 months or longer without replacement. This is based on less than 60 s of use per day and only requires rinsing with water daily [75]. The adoption of the proven Solo II vibrating mesh technology, meant Dance Biopharm acquired a durable aerosol generator at a relatively affordable cost due to low development cost [75]. A significant advantage of this Solo II vibrating mesh nebuliser is the ability to nebulise small doses from 0.050 to 0.225 mL of formulations, with less than 2 µL residual remaining in the device reservoir, which minimises drug waste [74].

Figure 2 outlines the performance of the Dance 501, using the volume median diameter (VMD) and the flow rate of the Solo II VMN [74], represented for each analysed device with a green triangle. Clinical trial studies report on the performance of the Dance 501 device, which produces aerosol particles in the range 3–6 µm, with an aerosol output rate of 0.2–0.6 mL·min^−1^ [76]. However, the effective size of the droplets in an aerosol should be 2–5 µm for deep lung penetration as detailed by [8]. This 1 µm in droplet size difference may seem negligible, but it results in a 72% increase in the mass of the droplet between 5 and 6 µm. This mass increase causes sedimentation in the upper airways. Optimum percentage respirable dose (%RD above 70%, highlighted in red) is achieved with a VMD of 3–3.5 µm and a flow rate of 0.4 mL·min^−1^. Aerosol particles <5 µm are subject to inertial impaction in the conducting airways. The droplet size is directly proportional to the orifice radius of the VMN device [11]. Attempts to reduce the orifice diameter to reduce droplet size result in a decrease in output flow rate to less than 0.2 mL·min^−1^ [74]. Smaller particle size leads to higher respirable dose (i.e., alveoli deposition). Low flow rates require longer dosing times and multiple breaths. User feedback reports that subjects required more breaths to inhale the dose completely, which is a less desirable treatment option [74]. For this study, an upper limit of five breaths for an insulin dose was set as a design specification of the device. Explored within this trial study [74] are the breathing manoeuvres to increase deep lung deposition of aerosol. A prior inhaler device developed by Aerogen, utilised a more conventional breathing manoeuvre [75], consisting of a five-second inhalation. Consisting of the first 4 s of aerosol inhalation followed by 1 s of aerosol-free “chase” air, and then a five-second hold. From this study, the reported bioavailability was sufficient [76], but based on the flow output rates of the Solo II aperture plate within the Dance 501, would require more than ten breaths, with breath-holding to deliver a 255 µL dose of insulin. This illustrates that the Solo II aperture plate design is not a suitable design for inhaled insulin. Outlined in red in Figure 2 is the 3–3.5 µm droplet size, which results in the highest % RD [74]. A volume flow rate of 0.4 mL·min^−1^ results in the highest % RD [74].

Statistical analysis was carried out using SPSS 20.0 software (IBM, Armonk, NY, USA). ANOVA, Shapiro–Wilk, and Wilcoxon tests were used. Group data were summarized using means and standard deviations. Differences between groups were evaluated by Mann–Whitney test. All tests were conducted at a 95% confidence level and significance level of *p* ˂ 0.05 [74].

## 5. Conclusions

For insulin-dependent type-I diabetics, continuous subcutaneous insulin infusion (CSII) pumps allow for effective and passive insulin therapy for type-I diabetes. For insulin-requiring type-II diabetics, CSII is considered to be excessive due to cost, surgical implantation, and the continuous functionality of the pump. Therefore, for insulin-requiring type-II diabetics, subcutaneous delivery of insulin is currently the most effective means of therapy available. However, subcutaneous insulin therapy has proved to be a burden on the lifestyle of people with diabetes. This burden leads to people with type-II diabetes being opposed to subcutaneous insulin and to cease use of subcutaneous insulin to control their blood glucose levels. With the cessation of insulin, blood glucose levels are controlled with lifestyle and dietary changes; this has shown to give poor and suboptimal control of blood glucose levels. Poor control of blood glucose leads to hyperglycaemia and associated health concerns. Therefore, a less invasive and passive alternative to insulin therapy is required for people with type-II diabetes. Alternative methods, such as inhaled insulin therapy have proved feasible. However, dry powder inhaler (DPI) devices have demonstrated irritative and poor performance in clinical trials. Irritative and poor performance have contributed to poor clinical approval. Many DPIs were designed and trialled, but only the Afrezza TI DPI device obtained FDA approval and is currently commercially available. However, chronic coughing directly after the use of the Afrezza TI device has contributed to the discontinuation of use of the device and has not obtained mass appeal.

Vibrating mesh technology has the potential to passively deliver inhaled insulin for effective glycaemic control to treat diabetes. Currently, the Dance 501 VMN inhaler is being developed for this purpose. However, clinical trials have shown that the Dance 501 device requires multiple breaths to administer insulin to achieve euglycaemia. To effectively achieve euglycaemia in a single breath with a VMN device requires the development of a new vibrating mesh component, which provides a higher flow rate while maintaining the volumetric median droplet size.

## Figures and Tables

**Figure 1 ijerph-17-05795-f001:**
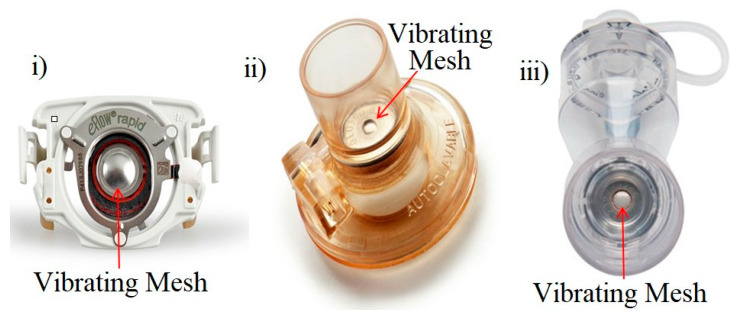
Current vibrating mesh technology. (i) PARI eFlow rapid vibrating mesh nebuliser [68]. (ii) Aerogen Aeroneb Pro vibrating mesh nebuliser [66]. (iii) Aerogen Solo II vibrating mesh nebuliser [66].

**Figure 2 ijerph-17-05795-f002:**
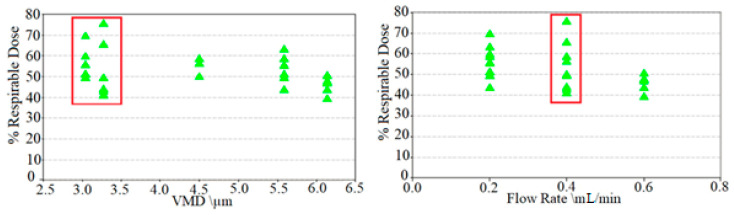
Volume median diameter (VMD) and volume flow rates of the Dance 501 collected distal to the Alberta throat with a simulate inspiratory flow rate of 10 L·m^−1^. The VMD is sufficient; however, the flow rate is inadequate and required multiple breaths and breathing techniques to administer a dose. The volume median diameter (VMD) to percentage respirable dose (%RD) of the Dance 501.

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
