# Peer review of "A Review: The Prospect of Inhaled Insulin Therapy via Vibrating Mesh Technology to Treat Diabetes"

_ijerph, 2020, doi:10.3390/ijerph17165795_

Round 1

Reviewer 1 Report

Authors proposed to investigate the potential treatment of diabetes with a wet insulin aerosol. The objectives are clear and cohesive. This review fits within the scope of the journal providing scientific and technical information in the field. Even though there are a few other publications covering the same aspects (e.g. PMID: 28571200, PMID: 20234779, PMID: 17594734, and more), this review brings a more technical approach to the field. 

Below I have some comments: 

1) Minor spell check required (e.g. lines-27-28 "Traditionally inhaled aerosol therapy has is used...")

2) Line 137-139 authors stated that insulin pumps are only used for T1D and it is not appropriate for T2D. This piece of text has no reference so I assume it is authors' own opinion. I suggest authors to reformulate this sentence and/or add appropriate citations that corroborate with authors' opinion since there are a few case reports demonstrating the use of pumps in type 2 diabetes with extreme insulin resistance and poor glycemic control. This also applies for the Conclusions section

3) Figure 2 outlines the VMD and volume flow rates if the Dance 501. It is unclear how the statistics were performed on the original paper and the values seem quite identical. I suggest authors to clearly demonstrated how the analysis were performed for clarification's sake 

Author Response

Lines 27-28 ;       Minor typographical corrected

Lines 137- 139 ; The syntax of the sentence has been amended to read “Insulin pumps are primary used for type-I diabetes, as the patient is reliant upon continuous insulin administration [32]; continuous insulin infusion is not required by people with type-II diabetes [31]. Thus, there is a desire to develop new forms of delivery devices to administer insulin to treat type-II diabetes [27].”

                                Providing a reference to the use of continuous insulin infusion pumps and rational to the desire to develop administration methods more suited to treatment of type-II diabetes.

Figure 2;               Gave further detail on  Shapiro-Wilk and Wilcoxon test, used to give statistical significance to the study.

Reviewer 2 Report

Cunningham et al have written a very nice review article on the prospects of inhaled insulin therapy. They have shown that vibrating mesh technology can be very effective to treat insulin requiring diabetics but with slight modifications. The manuscript is well designed but needs a few improvements as outlined below.

1- Line 31-34, Only in recent…………… systemic applications; looks very long and cumbersome for the readers and can be shortened and simplified.

2- Line 65, Looks like there is a space issue.

3- Line 120, To defined should be to define and please correct the syntax of this sentence.

Author Response

Lines 31-34;        Sentence structure simplified to read: “Only in recent decades has pulmonary drug delivery for systemic applications (i.e. conditions outside the respiratory system) been viable, due to increased efficacy of aerosol therapy [4].”

Lines 65;               Space removed.

Line 120;              Sentence and syntax restructured to read; “To define the aerosol flow rate and droplet size to treat type-II diabetes effectively using a vibrating mesh nebulisers.”

Reviewer 3 Report

A well-written summary on the options of lung administration of insulin. I have not found any serious concern about the manuscript.

Let me ask if you have encountered possible neoplastic complications of inhaled insulin in the literature. In connection with previous attempts of its nasal application, I read that one of the reasons why it was terminated was mitogenic effects on the nasal mucosa.

Only detail is a misuse of certain abbreviation - not abbreviated when first stated in the text (T2DM, FDA, TI)
Line 28 - "has is used"?
Line 115 - „in“ instead of „is“?
Line 277 - the line repeats abovementioned information

Author Response

Previous attempts used a dry powder insulin. This paper is focused on the oral inhalation of a wet insulin aerosol to mimic the humidity of the upper airways. Oral inhalation offers less resistance in the airways when compared to the nasal respiratory route, and is thus more desirable for systemic drug delivery.

Line 28 ;               Minor typographical corrected to read “has been used”

Line 115;              Minor typographical corrected to read “The fluid layer at the cell surface in the respiratory airways decreases from 8 µm to approximately 70 nm, which directly correlates with a decrease in cell thickness [19].”

Line 277;              Repetitive sentence removed.

This manuscript is a resubmission of an earlier submission. The following is a list of the peer review reports and author responses from that submission.